# An Analysis of the Biotin–(Strept)avidin System in Immunoassays: Interference and Mitigation Strategies

**Amy H. A. Balzer \* and Christopher B. Whitehurst \***

Department of Pathology, Microbiology, and Immunology, New York Medical College, Basic Medical Science Building, 15 Dana Rd., Valhalla, NY 10595, USA

\* Correspondence: amybalzer13@gmail.com (A.H.A.B.); cwhitehu@nymc.edu (C.B.W.)

**Abstract:** An immunoassay is an analytical test method in which analyte quantitation is based on signal responses generated as a consequence of an antibody–antigen interaction. They are the method of choice for the measurement of a large panel of diagnostic markers. Not only are they fully automated, allowing for a short turnaround time and high throughput, but offer high sensitivity and specificity with low limits of detection for a wide range of analytes. Many immunoassay manufacturers exploit the extremely high affinity of biotin for streptavidin in their assay design architectures as a means to immobilize and detect analytes of interest. The biotin–(strept)avidin system is, however, vulnerable to interference with high levels of supplemental biotin that may cause elevated or suppressed test results. Since this system is heavily applied in clinical diagnostics, biotin interference has become a serious concern, prompting the FDA to issue a safety report alerting healthcare workers and the public about the potential harm of ingesting high levels of supplemental biotin contributing toward erroneous diagnostic test results. This review includes a general background and historical prospective of immunoassays with a focus on the biotin–streptavidin system, interferences within the system, and what mitigations are applied to minimize false diagnostic results.

**Keywords:** immunoassays; interference; biotin; avidin; streptavidin

## 1. Introduction

Immunoassays have made an impact on the field of modern clinical medicine among the ranks of the discoveries of X-rays, penicillin, vaccines, DNA, and the human genome project [1]. They have made it possible to measure minute quantities of virtually any interesting biomolecule with high sensitivity and specificity, even in the presence of hundreds of thousands of other molecules [2]. Immunoassays gain their exceptional specificity and sensitivity from antibodies, which have the ability to bind to an enormously wide range of biological and man-made analytes (e.g., chemicals, biomolecules, cells, and viruses) with extraordinary affinity and avidity [3]. The quantitative measurement of this antibody–analyte association is significant as it provides critical information for deriving a fundamental understanding of molecular function. Predictive and mechanistic models of these binding associations [4] have provided physicians with a basis to scientifically define physiological and pathophysiological states to aid in the diagnosis, prognosis, and treatment of disease in their patients [2]. These models have also provided researchers biotechnological tools to further advance their understanding in subject areas such as molecular mechanism elucidation [5], drug discovery [6], and the evaluation of food and environment safety [7].

The combination of an easily detectable signal and a protein which binds with high affinity to an analyte is principle to all immunoassay methods [8]. What distinguishes immunoassays from other assays is their employment of an immune complex, relying on the reaction between an antibody and an antigen for the measurement of an unknown concentration of analyte in a sample [9]. Immunoassays can be broadly defined as quantitative and qualitative techniques of measuring analytes using antibody–antigen interactions [6],

coupled with a detection system that generates a signal response from a label (e.g., radioisotopic, enzymatic, fluorescent, or chemiluminescent [10]). Once the immune complex is formed, the label activity is measured and interpolated using a standard curve [6] that represents the measured signal as a function of the concentration [11].

Immunoassays are routinely used in laboratory diagnostics [12] due to their utilization of highly specific antibodies that exhibit remarkable diversity to selectively target a vast variety analytes of interest and measure their concentrations in a sample [13]. Antibodies are the key reagents on which the success of the immunoassay depends [6], relying on their ability to bind to a specific area of an antigen called an epitope [11]. Antibodies are glycoproteins secreted by plasma cells in response to antigen invasion, acting as one of the principle effectors of the adaptive immune response designed to neutralize and/or eliminate antigens [6]. Antigen interaction is central to an antibody's natural biological function. To bind to a diverse spectrum of antigens, antibodies require sequence diversity, which is achieved using a combination of different heavy and light chains to produce the antibody's binding site, called the complementarity-determining region (CDR) [14]. The widespread use of immunoassays is attributed to their antibodies' inherent specificity and high sensitivity, offering the ability to measure the concentrations of a vast range of small and large analytes, including macromolecules, drugs, metabolites, and/or biomarkers [6].

## 2. Immunoassay Design

While the design possibilities of immunoassays are limitless, the basic principle is based on a binding reaction between a finite number of binding sites for an unknown amount of analyte in the sample and a fixed amount of label to detect the analyte captured [6]. The design choice for an immunoassay is dependent on the nature of the analyte, labeling chemistry, and analytical parameter required from the assay (e.g., sensitivity, precision) [11]. They can be generally classified as homogeneous or heterogeneous, direct or indirect, and non-competitive or competitive.

### 2.1. Homogeneous vs. Heterogeneous Immunoassay Design

Homogeneous and heterogeneous immunoassays differ in that homogeneous immunoassays do not require a separation step. Homogeneous immunoassays can be performed directly in solution with minimal technique, enabling one-step, rapid detection of analytes, while heterogeneous immunoassays require multiple time-consuming incubation and washing steps [15]. Homogeneous immunoassays do not require a separation step since their detection signal is produced directly by the function of immunochemical binding. Heterogeneous immunoassays' detection signals are dependent on the location of an inert label and thus require the separation of free and bound (inert) labels [16]. Homogeneous immunoassays' detection systems require advanced instruments and specialty labels to measure signal activity [17]. Examples of these methods include particle agglutination using gold nanoparticles, fluorescence polarization using FRET (Förster Resonance Energy Transfer), and enzyme activity using EMIT (Enzyme-Multiplied Immunoassay Technique) [16]. Many homogeneous immunoassays rely on FRET techniques (Figure 1a) [18], which are based on a distant-dependent energy transfer between a donor and an acceptor label upon an antibody–antigen interaction. Immune complex binding events induce FRET by changing the distance between the excited donor fluorophore and a proximal ground state acceptor fluorophore [15]. In theory, homogeneous immunoassays are more sensitive than heterogeneous immunoassays as the separation and washing steps of heterogeneous immunoassays can be prone to error and reverse weak binding reactions. However, because the sample constituents of homogeneous immunoassays are not removed, non-specific signal can occur [16]. As a consequence, heterogeneous immunoassays are the preferred choice due to their higher sensitivity and lower detection limits.

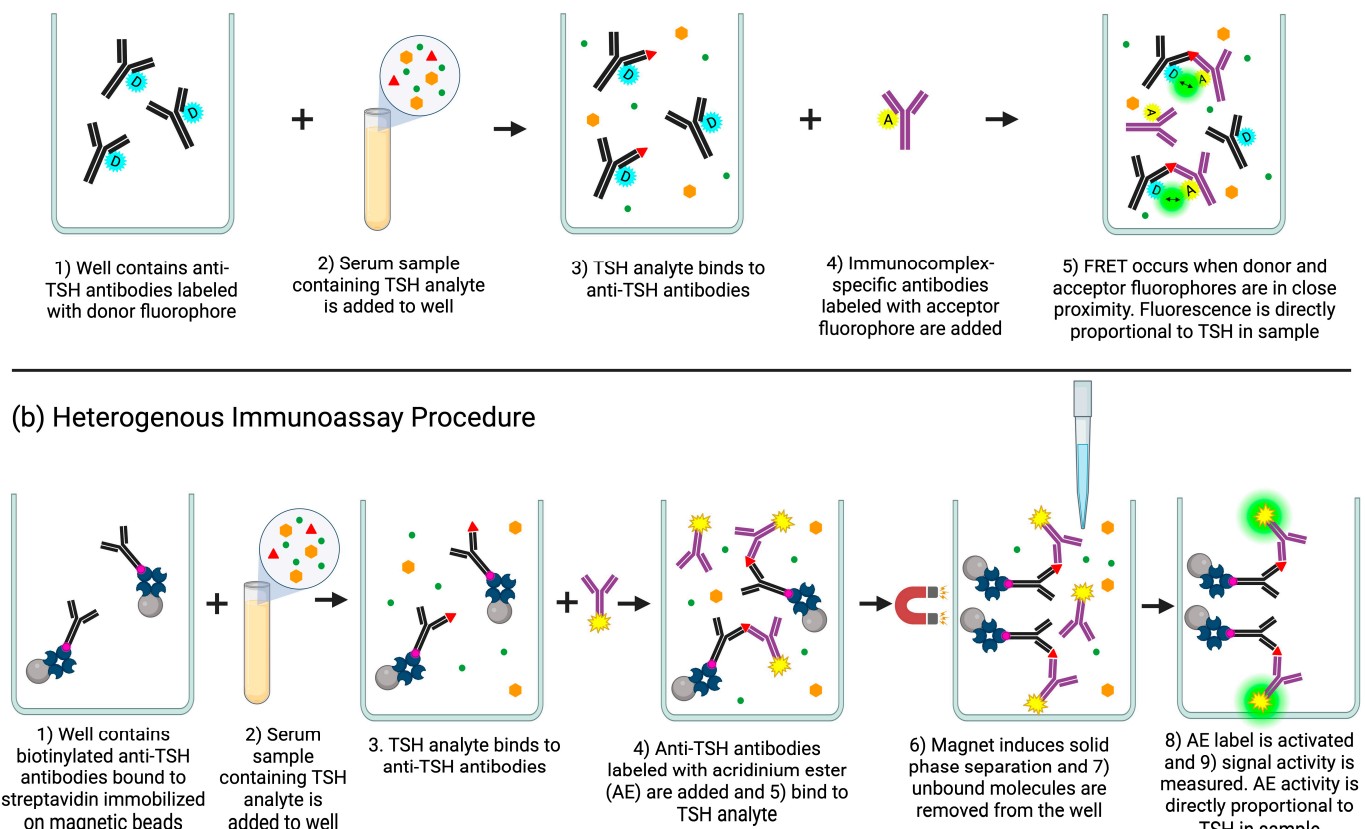

**Figure 1.** Procedural differences of homogeneous assays and heterogeneous assays for the quantitation of thyroid stimulating hormone (TSH) in a sample. Created using BioRender.com accessed on 29 October 2023. (**a**) Homogeneous immunoassay procedure. (1) Anti-TSH antibodies conjugated with a donor fluorophore (D) are added to a well (2) followed by the addition of sample containing TSH analyte. (3) TSH analyte from the sample binds to the anti-TSH antibodies. (4) Immunocomplex-specific antibodies conjugated with an acceptor (A) fluorophore are added and (5) bind to the TSH–anti-TSH immunocomplex, bringing the donor and acceptor fluorophores in close proximity, inducing FRET. Fluorescence is directly proportional to TSH present in sample. (**b**) Heterogeneous immunoassay procedure. (1) Biotinylated anti-TSH antibodies bound to immobilized streptavidin on magnetic beads are added to a well (2) followed by the addition of sample containing TSH analyte. (3) TSH analyte from the sample binds to the anti-TSH antibodies. (4) Anti-TSH antibodies labeled with acridinium ester (AE) are then added and (5) bind to TSH analyte complex. (6) A magnet induces solid phase separation by immobilizing the magnetic beads with the bound immunocomplex while (7) unbound molecules are washed and removed from the well. (8) An addition of reagents induces a chemical reaction activating the AE label and (9) AE activity is measured. AE activity is directly proportional to the TSH present in sample.

In heterogeneous immunoassays (Figure 1b), the immune complex is immobilized on a solid support (e.g., well plate or magnetic bead) and unbound molecules are separated and washed away [19]. Fixation of an antigen, antibody, or protein to a solid support serves to immobilize the target analyte, ideally without disturbing the cellular architecture and allowing maximum binding access to any targeted cellular component of the analyte [20]. Examples of immobilization methods include passive adsorption (e.g., hydrophobic, van der Waals, and pi–pi interactions), crosslinker-mediated (e.g., glutaraldehyde, carbodiimide, maleimide, and hydrazide), and site-directed capture (e.g., affinity tag biotin–(strept)avidin, enzyme–substrate, immunoglobulin-binding protein A and G). Immobilization via biotin–

(strept)avidin is one of the most popular approaches thanks to its good stability, high efficiency, high specificity, and high binding affinity [21]. Analyte quantitation is performed by measuring the label activity (e.g., radioisotopic, enzymatic, fluorescent, or chemiluminescent [10]) bound to the immobilized immune complex [19]. In the chemiluminescence technique, after the separation and removal of the unbound label, a reagent is added to initiate a chemical reaction that induces the transition of the label molecule's electrons from a ground state to an excited state. The return of the electron to the ground state emits a photon of light that is used to measure the analyte concentration [22].

## 2.2. Direct Immunoassay vs. Indirect Immunoassay Design

Direct (Figure 2a) and indirect (Figure 2b) immunoassays differ in the number of antibodies integrated into the immobilized immune complex. Direct immunoassays employ detection methods using one labeled primary antibody that binds directly with the analyte. Indirect immunoassay detection methods employ two antibodies, where one antibody (primary) binds directly to the analyte and a second antibody (secondary) conjugated with a label indirectly binds to the analyte by binding to the primary antibody [11]. Although the direct design is faster at generating results, the indirect method is more frequently used in immunoassays as it offers higher sensitivity, greater signal amplification, and the ability to detect several analytes in the same sample [20]. Direct detection methods are more commonly used for the immunohistochemical staining of tissues and cells [11]. Indirect immunoassay techniques are, however, prone to cross-reactivity with the endogenous immunoglobulins found in the sample matrix. To prevent this, the secondary antibodies are derived from a different species than that of the sample. For greater signal amplification, polyclonal antibodies can be used as they can recognize multiple epitopes of the primary antibody. Signal amplification offers increased binding and signal levels [20], increasing the sensitivity of measuring minute quantities of the analyte [23].

## 2.3. Competitive Immunoassay vs. Non-Competitive Immunoassay Design

The general distinction between competitive and non-competitive immunoassays lies in the relationship between the label activity and analyte concentration in a sample. In competitive designs, the label activity is inversely proportional to the analyte concentration, while in non-competitive designs, the label activity is directly proportional to the analyte concentration. Competitive immunoassays rely on the competition between the analyte from the sample and a labeled analyte (or analogue) from the reagent for a limited number of binding sites (Figure 2d). As the concentration of analyte in the sample increases, the availability of binding sites for the labeled analyte decreases, resulting in a reduction of signal activity [24]. To quantify the antibody, the labeled antibody competes with the antibody from the sample for binding to a limited number of immobilized capture antigens [25]. For antigen quantification, the labeled antigen competes with the antigen in the sample for binding to the immobilized capture antibodies. Competitive designs are often used when the analyte is small or when a matched pair of antibodies to the analyte does not exist. A drawback of competitive immunoassays is that they are not as sensitive as non-competitive immunoassays as they are more prone to sample matrix effects [25].

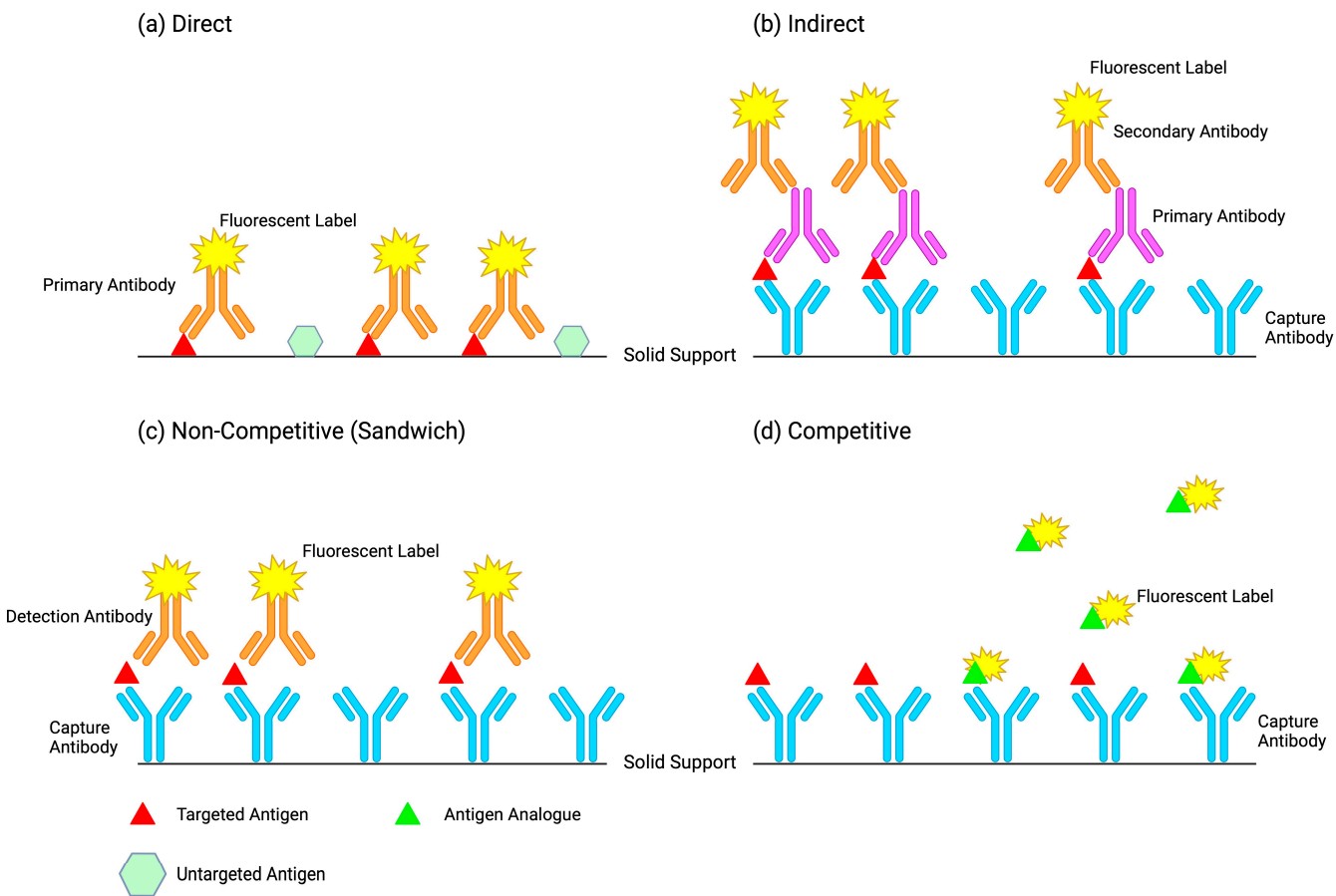

**Figure 2.** Design possibilities for quantifying thyroid stimulating hormone (TSH) in serum using heterogeneous methods. Created using BioRender.com accessed 29 October 2023. (**a**) Represents a direct design where the target antigen TSH is immobilized on the solid support along with other untargeted antigens from the sample matrix. An anti-TSH antibody labeled with a fluorescent probe binds specifically to the target TSH antigen only. Fluorescence is directly proportional to the amount of TSH present in sample. (**b**) Represents an indirect design where a primary antibody (mouse anti-TSH antibody) binds to TSH bound to a capture antibody (human anti-TSH antibody) immobilized on a solid support. A secondary antibody (anti-mouse antibody) labeled with a fluorescent probe binds to the mouse anti-TSH antibody. Fluorescence is directly proportional to the amount of TSH in the sample. (**c**) Represents a sandwich design where TSH is bound ("sandwiched") between an anti-TSH capture antibody and an anti-TSH detection antibody. Capture antibody is in excess. Fluorescence is directly proportional to the amount of TSH present in sample. (**d**) Represents a competitive design where TSH in the sample competes with a TSH analogue labeled with a fluorescent probe for binding to an immobilized anti-TSH antibody. Detection analogue is in excess. Fluorescence is inversely proportional to the amount of TSH in the sample.

Non-competitive immunoassays (also known as "sandwich" immunoassays) use two antibodies, one for capture and one for detection (Figure 2c). A capture antibody highly specific to the antigen of interest is immobilized on a solid support. The sample containing the antigen is then introduced, along with a second antibody specific to the antigen that has been conjugated with a label for detection. The detection and capture antibodies are both highly specific, but bind to different epitopes of the antigen. As a result, the antigen is "sandwiched" between the two antibodies. If the immunoassay detects antibody in the sample, the antibody is "sandwiched" between two antigens, two antibodies, or a combination of the two [25]. As the concentration of analyte in the samples increases, the availability of binding sites to the label increases, resulting in an elevation of signal activity.

## 3. The Biotin–(Strept)avidin System

In complex matrices such as serum or plasma, capturing an analyte of interest with high specificity and sensitivity is crucial [21]. In addition to the antibody–antigen system, immunoassays have employed other high-affinity systems to further enhance their specificity (Table 1). A prevalent, well-characterized immobilization design system is the biotin–(strept)avidin interaction [26]. The biotin–(strept)avidin interaction is considered to be one of the most specific and stable non-covalent interaction, whose dissociation constant ($K_D$) is about $10^3$ to $10^6$ times higher than an antigen–antibody interaction [27]. Its high affinity is principally useful for isolating and amplifying the signal, which increases the ability for the detection of very low concentrations of analyte while decreasing the number of steps required for measurement, allowing for a more rapid quantitation of analyte [28]. The biotin–(strept)avidin system offers enormous advantages over other covalent and non-covalent interactions, which include amplification of weak signals, efficient operation, robustness, and astonishing stability against manipulation, proteolytic enzymes, temperature and pH extremes, harsh organic reagents, and other denaturing reagents. Since the biotin–(strept)avidin interaction is one of the strongest known non-covalent interactions in nature, avidin and its analogues have therefore been extensively used as probes and affinity matrices for a wide variety of applications in the field of biotechnology, such as biochemical assays, diagnostics, affinity purification, and drug delivery [27]. Table 1 demonstrates the significantly greater binding affinity of biotin–(strept)avidin interactions when compared to other systems.

**Table 1.** Comparison of binding interaction affinities.

| System | Affinity $K_D$ | Reference |
|---|---|---|
| Biotin–(strept)avidin | $10^{-14}$–$10^{-15}$ | [29] |
| His$_6$-tag–Ni$^{2+}$ | $10^{-13}$ | [30] |
| Monoclonal antibodies | $10^{-7}$–$10^{-11}$ | [31] |
| RNA–RNA binding protein | $10^{-9}$ | [4] |
| Nickel–nitrilotriacetic acid (Ni$^{2+}$–NTA) | $10^{-13}$ | [32] |
| Dinitrophenol (DNP)-anti-DNP | $10^{-8}$ | [33] |
| Biotin–anti-biotin antibody | $10^{-8}$ | [34] |

### 3.1. History

The first immunoassay was developed in 1959 by Solomon Berson and Rosalyn S. Yalow for the detection of insulin in patient blood samples [1]. Their previous work on insulin metabolism helped them recognize that antibodies can be used to study this hormone [2]. They discovered that the competition of endogenous human insulin with radioactively labeled ($I^{131}$) tyrosine residues of bovine insulin in binding to the anti-insulin antibodies in the sera of guinea pigs immunized with bovine insulin [35] provided a basis for a sensitive and specific assay to quantify insulin in patients [2]. This first radioimmunoassay won the Nobel Prize in Medicine in 1977 and paved the way for the development of other types of immunoassays [1]. In 1971, Eva Engvall and Paul Perlmann pioneered the enzyme-linked immunosorbent assay (ELISA) by immobilizing antigens on a solid support to measure the concentration of antibodies in a sample using an enzyme-linked anti-immunoglobulin detection antibody. Simultaneously, B.K. van Weeman and A.H. Schuurs developed an assay with the same principles quantifying an antigen, rather than an antibody [13]. Since then, the ELISA method has become a standard of routine laboratory research and diagnostic methods worldwide [36].

In 1979, Jean-Luc Guesdon, Therese Ternynck, and Stratis Avrameas, a team from the Institut Pasteur in France, developed the first ELISA employing the avidin–biotin system. They created two different methods by conjugating biotin to antibodies, antigens, and enzymes, which then binds to avidin to detect, immobilize, and quantify antibodies and/or antigens of interest. In the Bridged Avidin–Biotin (BRAB, Figure 3a) method, the antigen from the sample is "sandwiched" between an immobilized capture antibody and a biotin-

labeled antibody. After a washing step eliminating the unbound biotin-labeled antibody, avidin is added and binds to the biotin label on the immune complex. A second washing step occurs to eliminate the unbound avidin, followed by the addition of a biotin-labeled enzyme that binds to the immobilized avidin. Lastly, a third washing step removes the unbound biotin-labeled enzyme, and the enzyme activity associated with the presence of antigen in the sample is measured. In the Labeled Avidin–Biotin (LAB, Figure 3b) technique, the antigen from the sample is bound to an immobilized antibody and a biotin-labeled antibody, just like in the BRAB technique. However, in the LAB technique, the avidin is pre-labeled with the enzyme, eliminating the need for an extra step [37].

**Figure 3.** Methods of BRAB (**a**) and LAB (**b**). Created using BioRender.com accessed 29 October 2023.

The applications and designs of the biotin–streptavidin system are vast, giving researchers the ability to perform a variety of tasks including identification, localization, and quantitation of the target analyte. The system allows for an indirect interaction between two biomolecules, preserving the natural binding properties of the antibodies and antigens. Biotin has the ability to be conjugated to a wide variety of biomolecules without altering the interaction of the biomolecule with its target ligand [38]. Biotin's relatively small size (240 Da), flexible valeric side chain [39], and ease of conjugation [27] make it well suited to protein labeling [40].

### 3.2. Biotin

Biotin was identified from a seemingly simple observation; consuming raw egg whites leads to toxicity, and this toxicity can be prevented using an unknown substance in the egg yolk [41]. In 1916, W. G. Bateman of Yale University made the casual observation that raw egg whites in the diet of animals had a toxic effect. Then, 11 years later, Margaret A. Boas at the Lister Institute of Preventative Medicine in London came across the same phenomenon. By feeding raw egg whites to rats as the source of protein in their diet, she observed that they developed dermatitis and hemorrhages of the skin, loss of hair, paralyzed limbs, lost considerable weight, and eventually died. Cooking the egg whites did not elicit toxic symptoms and its effects could be alleviated or prevented by the consumption of food containing biotin. Egg whites contain the toxic protein avidin, which is neutralized via its binding to biotin (a component of egg yolks), preventing its absorption and toxic effects. Biotin was then discovered in 1936 by Fritz Kögl and B. Tönnis from the University of Utrecht, Netherlands, who were looking to find a source of a minute quantity of an

unknown compound present in a charcoal fraction of "brewer's wort" (an extract of ground malt) that was necessary for the growth of yeast cultures. Kögl found that egg yolks contained the highest concentration he could find of this substance. In a series of 16 tedious steps from 550 pounds of dried duck egg yolks, Kögl and B. Tönnis succeeded in isolating the active principle of this fraction, which ultimately turned out to be biotin [42].

Biotin (vitamin H, B$_7$ [43], coenzyme R [29]) is a water-soluble B-complex vitamin and essential coenzyme for five human carboxylases (acetyl–CoA carboxylase (ACC) 1 & 2, pyruvate carboxylase (PC), propionyl–CoA carboxylase (PCC), and 3-methylcrotonyl–CoA carboxylase (MCC)) [44]. It is composed of a tetrahydrothiophene ring, a valeric acid side chain, and a ureido (tetrahydroimidizalone) ring [45] that functions as the carrier for CO$_2$ [41]. Biotin has a well-studied role as a covalently bound coenzyme that catalyzes critical steps in the metabolic pathways of gluconeogenesis, fatty acid synthesis and oxidation, and amino acid catabolism. But more recently, it has been revealed that biotin has noncarboxylic functions involved in cell signaling, the epigenetic regulation of genes and the chromatin structure, and immune response [44]. Biotin binds to avidin with a K$_D$ in the order of ~$10^{-15}$ M and binds to streptavidin with a K$_D$ in the order of ~$10^{-14}$ M [29], forming the strongest known noncovalent ligand–protein interaction, achieving picomolar and even femtomolar affinities [39]. Binding occurs very rapidly and remains undisturbed by pH and temperature extremes, organic solvents, and other denaturing agents, making this a stable and ideal system to use in immunoassays [29].

### 3.3. (Strept)avidin

Avidins are a family of biotin-binding proteins that are found in both eukaryotic and prokaryotic species. The first known avidin was isolated from chicken (*Gallus gallus*) egg whites in 1941 [46] by researchers Robert E. Eakin, Esmond E. Snell, and Roger J. Williams. They confirmed that the constituent in raw egg whites capable of inactivation using biotin in vitro [47] and eliciting toxic effects when fed to animals [48] is a protein called avidin [49], named for its avidity with biotin (avidity + biotin) [50]. The first bacterial avidin, streptavidin, was isolated from the antibiotic-secreting *Streptomyces avidinii* bacteria in 1964. Since then, several new avidins have been experimentally verified from both eukaryotic and prokaryotic species [46]. Several genetically and chemically engineered avidins and their analogues have also been studied to advance our knowledge about the functional and structural characteristics of avidins, potentially leading to more successful applications [27].

Avidin and streptavidin are functional and structural [51] tetrameric glycoprotein [27] analogues that contain four biotin-binding sites [51]. Although the functional properties and the quaternary and tertiary structures of these two proteins are well conserved, their primary structures have low similarity (~30%) [46]. Each of the four monomers contains eight antiparallel β-strands that form a β-barrel. At one end of each β-barrel is a biotin-binding site that comprises several aromatic residues [39] and a tryptophan (Trp) residue from the neighboring subunit. The residues from the biotin-binding site bind with the ureido group of biotin using hydrogen bonds and van der Waals interactions [52]. A flexible loop of 12 and 8 amino acids of avidin and streptavidin, respectively, closes the binding pocket on top of biotin, which contributes significantly to its binding strength [5]. The tryptophan from the adjacent monomer acts as a hydrophobic lid for the binding pocket [45], making it one of the most important amino acid residues responsible for the tight biotin interaction [53]. Two of the monomers lie parallel to each other, forming a dimer with an extensive interface, and the two dimers associate, forming a weaker interface. Interestingly, the weaker dimeric interface appears to be important to the high affinity as mutations either increase or decrease the protein's affinity to biotin [5]. A mutation in Trp-110 and Trp-120 in avidin and streptavidin, respectively [45], results in only monomers, which significantly reduces the affinity (K$_D$~$10^{-7}$ M). As new applications of (strept)avidin–biotin complexes are explored, more avidin derivatives will surely continue to emerge [5].

Chivers et al. engineered an avidin derivative called traptavidin, which exhibits stronger biotin-binding due to two mutations [39] that stabilize the binding pocket loop by reducing its flexibility [54] and reducing its conformational change upon biotin binding [55]. Additionally, researchers have engineered other avidin-derived proteins in attempt to circumvent the pitfalls of avidin [34], such as a reduced binding affinity when biotin is conjugated to a protein [29], non-specific binding, and immunogenicity [27]. Although avidin has a stronger biotin-binding affinity, its basic isoelectric point (pI ~10) makes it prone to nonspecific adsorption of negatively charged molecules [56] and surfaces such as cell membranes or silica substrates at physiological pH [51]. Another feature contributing to the high degree of nonspecific adsorption is its carbohydrate groups, which contain four mannose and three N acetyl glucosamine residues in each monomer. Carbohydrate-binding molecules in the sample matrix can bind specifically to the carbohydrate groups of avidin, limiting its use. Streptavidin, a nonglycosylated avidin, and neutravidin, an engineered commercially available deglycosylated form of avidin, lacks the four mannose and three N acetyl glucosamine residues in each subunit of avidin [34]. The absence of the carbohydrate moieties make the pI of streptavidin (pI~5–6) and neutravidin (pI~6.3) more acidic, reducing the nonspecific binding [51] without significantly affecting biotin affinity [34]. Since the biotin-binding affinity of streptavidin and neutravidin in solution is similar, they are often used interchangeably. It is assumed that streptavidin and neutravidin have interchangeable functionalities due to their similar biotin-binding abilities. However, a study by Sut et al. concluded that the adsorption rates of streptavidin and neutravidin with a biotinylated supported lipid bilayer (SLB) interface differed in a pH-dependent manner, and their attachment to these biotinylated SLB interfaces is very different in terms of attachment kinetics, adlayer properties, and pH sensitivity. This tell us that depending on the application of use of the system, the pI and other relevant interfacial forces in the system must be taken into account when choosing between avidins for a design [56]. Predicting what product is most appropriate for a given application is complicated by the fact that there are numerous and varied formats of avidins available with no universal method of measurement for biotin binding, making it challenging to accurately compare expectations among products [57]. Table 2 displays the properties of common avidin derivatives.

**Table 2.** Properties of avidin derivatives.

| | Origin | MW | pI | Reference |
|---|---|---|---|---|
| Avidin | *Gallus gallus* egg white | ~66–69 kDa | ~10 | [27] |
| Streptavidin | Bacterium *Streptomyces avidinii* | ~56 kDa | ~5–6 | [27] |
| Neutravidin | Deglycosylated avidin | ~60 kDa | ~6.3 | [27] |
| Traptavidin | S52G, R53D mutant of streptavidin | ~56 kDa | ~5.1 | [54] |
| Bradavidin II | *Bradyrhizobium japonicum* | ~58.4 kDa | ~9.6 | [27] |
| Tamavidin 2 | Mushroom *Pleurotus cornucopiae* | ~ 60.9 kDa | ~7.4 | [58] |

## 4. Biotin Interference

Despite their great performance characteristics, immunoassays are prone to interferences that may alter tests results, putting patients at risk for misdiagnosis [12]. All biotin–(strept)avidin-based immunoassays are susceptible to interference with biotin, but the degree of risk for patient misdiagnosis can vary considerably between immunoassays [40]. The impact of the observed interference is dependent on the concentration of biotin, the analyte of interest [59], and the architecture of the affected assay causing falsely low (non-competitive) or falsely elevated (competitive) results [40]. Excess biotin from the sample can inhibit immune complex separation by binding to the binding sites of (strept)avidin reserved for the capture or detection of the analyte [60]. Non-competitive immunoassays display an interference in which, in the presence of high concentrations of biotin, the excess biotin saturates the immobilized streptavidin-binding sites and prevents linking with the analyte–antibody immunocomplex, leading to falsely low results. In competitive immunoassays, the excess biotin binds to the immobilized streptavidin on

the solid phase and prevents the binding of the endogenous analyte from the sample and the labeled detection analyte, leading to a falsely decreased signal and thus falsely high results. Non-competitive assays are in theory less vulnerable because the analyte-binding sites are in excess, whereas competitive assay analyte-binding sites are limited [61]. As a result, falsely elevated or suppressed results can lead to erroneous results that mimic pathological conditions. There have been reported cases of patient misdiagnosis of vitamin D intoxication, hyperandrogenism, testosterone-producing tumors, and hypercorticism linked to high biotin intake [60].

While the biotin–streptavidin system is extensively used in immunoassays, reports of biotin interference have until recently been relatively uncommon [62]. The combination of the growing utilization of the biotin–(strept)avidin system and the rising popularity of both therapeutic and non-therapeutic use of supraphysiological doses of biotin has led to more frequent reports of erroneous diagnostic results [60]. Unusually high concentrations of circulating biotin have risen in occurrence due to the excessive intake of supplements containing biotin [63]. Biotin has gained commercial popularity for its claims on benefitting hair growth [43] since it is an essential cofactor for mitochondrial carboxylases in the hair root. In fact, biotin earned the name "vitamin H" for its role in the hair root after "Haar und Haut", derived from the German words for "hair and skin". Although a substantial amount of market advertising and social media publicity exists for the efficacy of biotin supplements in the improvement of hair quality, the only human health condition for which there is strong evidence of biotin therapeutic utility is for the treatment of biotin deficiency. The initial literature investigating the efficacy of biotin on hair quality dates back to 1965, in which 46 women were treated with unknown doses of biotin and observed for its "effects on hair roots". The researchers concluded that biotin supplementation did not produce a change in the "state of the hair roots" in any of the 46 women. Even though it was established as early as 1965 that biotin does not enhance hair quality, companies still promote biotin supplements directed at the improvement of hair quality [64].

The existing evidence supporting the efficacy of biotin supplements on hair growth is limited and therefore lacks proven benefits in healthy individuals [43]. There is, however, substantial evidence proving that excess consumption of biotin contributes to false immunodiagnostic results, enough for the Food and Drug Administration (FDA) to issue a warning. In 2017, and again in 2019 [65], the FDA issued a safety report to alert the public and healthcare workers about the potential harm of ingesting high levels of biotin. The FDA recommends that healthcare providers consider biotin interference as a potential source of error if a lab test result does not match with a patient's clinical presentation [66]. Although awareness of biotin interference among laboratory staff and clinicians is heavily implemented, many patients still consume high doses of biotin [65]. It is important for consumers to understand the significance of ingesting excess biotin and the consequences it may have on their lab results to help mitigate this healthcare issue [60].

Education serves as the primary safeguard for the prevention of biotin interference. Open communication between the laboratory and clinical providers is essential to ensure that clinicians and manufactures are aware of its potential impact. Patients should be advised to check the labels for biotin content in the supplements they are currently taking and disclose their use to their healthcare providers. Healthcare professionals should also actively question their patients about dietary supplement and biotin intake. However, the responsibility ultimately lies on the patient and their awareness of the importance of reporting biotin supplementation to their healthcare clinician [40]. The European Medicines Agency mandated a warning message that must accompany any biotin supplements containing a dose higher than 150 µg regarding the risk of erroneous diagnostic results [60]. Worldwide prevalence of high biotin supplementation is not well known but suggested to be county-dependent. A study quantifying the biotin concentration in plasma samples of patients in the emergency department from the United States demonstrated that 7.4% of these samples had a biotin concentration of above 10 ng/mL. Australia had 0.8%, the Netherlands had 0.2%, and the United Kingdom had no patients above a level of

2.5 ng/mL [60]. These are, however, isolated studies and should not be presumed representative of these entire countries' populations. As the rising prevalence of biotin interference is somewhat recent, further testing is necessary to assess its global and localized impact so that the proper mitigation strategies can be implemented. Estimates of biotin supplement usage are also varied as it is difficult to obtain accurate and reliable data on biotin supplementation. Healthcare professionals are less likely to be aware of patients taking biotin for hair/skin/nail improvement purposes, compared with those who are under physician care and are being prescribed biotin. In addition, patients may not even realize they are ingesting biotin or that it is relevant to report their supplement intake to their physician. The FDA only requires dietary supplement companies to be responsible for ensuring their products meet the standards in terms of safety and labeling and are not required to demonstrate product efficacy before launching it to the market, unlike product release in pharmaceuticals. Concerns are raised in regards to the variability in quality and safety of dietary supplements, as well as the FDA's ability to effectively regulate these products [40].

The Food and Nutrition Board of the Institute of Medicine recommends that adults have a daily dietary intake of 30 μg of biotin to maintain adequate health. Most healthy individuals following a western diet (35–70 μg/day) meet these requirements [67] as it is readily found in many foods (e.g., eggs, pork, cereals, leafy green vegetables) [40] and is produced by normal gut flora [43]. Since the recommended daily intake of biotin is relatively low and can be easily obtained from one's diet, biotin deficiency is extremely rare and is usually only observed in severely malnourished children or individuals with a biotinidase deficiency [40]. Despite the prevailing assumption that biotin deficiency is rare, there is mounting evidence of biotin deficiency in several physiological and pharmacological states. Apart from primary causes being neonatal biotinidase deficiency or inborn errors of biotin metabolism, biotin deficiency has also been associated with various factors such as protein–energy malnutrition, long-term parenteral malnutrition, anticonvulsant therapy, pregnancy, alcoholism, smoking, and aging. Pharmacologic uses of biotin have been limited to the treatment of conditions such as [40] biotin-responsive basal ganglia disease (100–300 mg/day), inherited metabolic diseases such as biotinidase (5–10 mg/day) and holocarboxylase synthetase deficiencies (30–40 mg/day), and multiple sclerosis (MS) (up to 300 mg/day) [29]. However, recent studies suggest that biotin supplementation can improve the treatment and management of diseases such as inflammatory bowel disease (IBD) [44], obesity [68], and type II diabetes [69].

For any diagnosis, it is important to obtain a complete picture when considering immunoassay test results in the context of a patient's clinical evaluation, as well as any diagnostic imaging. As much as 70% of medical decisions are based on laboratory results [70]. Interference should be expected when test results suggest a diagnosis inconsistent with clinical symptoms. It is recommended to comprehensively review medical records and medications for the presence of supplemental biotin or other interferents, and assess the direction of the suspected interference in the context of the assay format. Apparent anomalies unrelated to assay interferences may also occur due to other analytical issues inherent to the test or analyte itself (e.g., imprecision and biological variability) [40].

## 5. Mitigations

It is crucial to evaluate the potential impact of biotin interference on an assay's analytical performance [40]. Immunoassay manufacturers are faced with the challenge of developing new or modifying existing immunoassays to exhibit a biotin sensitivity that meets the FDA's desired biotin threshold of up to 3510 ng/mL as outlined in the Clinical and Laboratory Standards Institute (CLSI) guidelines. This biotin threshold level is three times higher than the highest physiological biotin concentration of 1160 ng/mL observed in patients with high-dose biotin intake for the treatment of MS [29]. Normal circulating biotin concentrations in a diet without supplementation are estimated to vary from 0.12 to 0.32 ng/mL or $0.60 \pm 0.15$ ng/mL, far lower than the required threshold [40]. Increasing

the biotin threshold makes an immunoassay less vulnerable to interference from high levels of supplemental biotin. Establishing a biotin threshold three times higher than the highest known concentration of circulating biotin ensures a relatively low probability of inaccurate tests results due to interference from biotin. As the CLSI guidelines are an essential requirement for global clinics and hospitals, manufactures must devote serious efforts to overcome this issue. While manufacturers have initiated the development of formats that are free from biotin interference, this change in immunoassay chemistry requires intensive clinical validation and regulatory approvals, which is a formidable undertaking [29].

Biotin interference thresholds can vary substantially between manufacturers. Some immunoassays exhibit greater sensitivity to biotin than others. Bowen et al. summarize the wide spectrum of biotin threshold levels found in some commercial assays with known interferences with biotin, and categorize them into groups targeting reproductive endocrinology, thyroid, and cardiac assays. They list biotin thresholds levels that demonstrate a ≤10% change in results due to interference ranging from as little as 2.5 ng/mL to as much as 10,000 ng/mL among these commercial kits. Notably, while this list is not exhaustive, there are no immunoassays for reproductive endocrinology (range 5.1–60 ng/mL) and thyroid assays (range 4.9–492 ng/mL) that meet the FDA's biotin threshold requirement of 3510 ng/mL [40]. As manufactures work to resolve this sensitivity issue, a need for effective biotin interference mitigation methods still remains. Current methods to verify the presence of biotin include conducting serial dilutions of the sample, retesting after biotin clearance, depletion of biotin from the sample, and/or repeat testing on an alternate platform [40].

### 5.1. Immunoassay Format Redesign

It is possible to eliminate biotin interference by eliminating the biotin–(strept)avidin system altogether and substituting it with another high-affinity system. One example is the fluorescein isothiocyanate (FITC)–anti-FITC system. This system is based on the reaction of a FITC label and an anti-FITC antibody for immobilization and detection. It has been described as being a sensitive alternative to the biotin–streptavidin system, achieving close (but lower) affinities with low non-specific binding [71]. Similarly, anti-2,4-dinitrophenol (DNP) antibodies can be used to measure analytes by detecting a DNP label with high affinity [72]. Capture antibodies/antigens can also be directly immobilized to a solid support via different chemistries. The covalent attachment of the capture protein to the solid support eliminates the need for immobilization with biotin and (strept)avidin. As a result, the probability of biotin binding to the immobilized immune complex is greatly reduced [29].

Another approach to reducing biotin interference is to ensure the biotin and (strept)avidin are pre-bound to each other before adding the sample. When the sample is introduced together with the biotinylated capture and/or detection antibodies, the biotin interference becomes more pronounced as the biotin in the sample competes with the biotinylated conjugates for binding to the immobilized (strept)avidin. This design reduces the number of steps in the manufacturing process, thus reducing testing time and costs, but dramatically decreases the biotin interference threshold. Assay formats that have the biotin–(strept)avidin pre-bound during the manufacturing process display considerably less interference with biotin [29].

Manufacturers can also increase the amount of (strept)avidin-binding sites to increase the biotin interference threshold. A study by Liu et al. determined that the essence of biotin interference is an insufficiency of streptavidin. They showed that by increasing the concentration of streptavidin-coated magnetic beads, thus increasing the number of streptavidin-binding sites, they could successfully neutralize biotin in a sample while maintaining precision and accuracy in both a competitive and noncompetitive design [73].

### 5.2. Use of Avidin Derivatives

Derivatives of avidin exhibit different binding capacities to biotin. Among its protein family, avidin demonstrates the strongest binding affinity to free biotin. However, when

binding to conjugated forms of biotin, streptavidin binds with higher affinity than avidin. These differences are thought to possibly be due to variations in the structure of the binding pocket loop [53]. Despite the low sequence homology of 30% identity (conserved amino acids) and 41% similarity (similar characteristics of amino acids), residues that are directly involved in biotin binding are mostly conserved. The avidin residues Trp-70, Phe-72, Phe-79, Trp-97, and Trp-110 and streptavidin residues Trp-79, Trp-92, Trp108, and Trp-120 are involved in the hydrophobic interactions with biotin [45]. Streptavidin has no analogous residue to Phe-72 in avidin, which may explain the weaker biotin interaction [53]. Based on these data, streptavidin appears to be less sensitive than avidin in regard to interference with free biotin. However, streptavidin maintains an extraordinarily strong affinity to biotin and their differences in response to biotin interference would likely be marginal.

Bradavidin II is a bacterial avidin found in *Bradyrhizobium japonicum*, a nitrogen-fixing and root-nodule-forming bacterium of the soybean plant. Bradavidin II was discovered to have a similar sequence homology to avidin and streptavidin of 38% and 32%, respectively [53]. The residues involved in the hydrophobic interactions with biotin are Trp-75, Trp-90, Phe-66, and Phe-42 [74]. Bradavidin II does not have a tryptophan analogous to the Trp-110 in avidin, which may explain its weaker ligand-binding capacity; however, a proline residue may functionally substitute tryptophan in bradavidin II. Bradavidin II binds to biotin with a $K_D$ of $10^{-10}$ M, a significantly less strong affinity than (strept)avidin [53]. Although there is no available evidence regarding the impact of biotin interference in an immunoassay containing bradavidin II, one can infer that proteins belonging to the avidin family possess such a strong affinity to biotin that interference with biotin will affect all avidin derivatives.

Several efforts have been attempted to use genetically modified avidin and biotin derivatives; however, many have led to a reduction in binding affinity [63]. Improvements in protein engineering have provided new possibilities to develop avidin derivatives. Avidin modifications aimed at reducing biotin binding have evolved from simple amino acid substitutions into more sophisticated changes, including chimeric avidins, topology rearrangements, and the stitching of non-natural amino acids into the active sites [75]. Suganuma et al. developed an enantiomer of streptavidin to be able to specifically bind to the enantiomer L-biotin instead of D-biotin as a potential candidate to reduce biotin interference [63]. Out of the eight different stereoisomers of biotin, only D-biotin is abundant in nature and exhibits biological activity [29]. Researchers developed the core streptavidin protein sequence using D-amino acids residues, instead of L-amino acid residues, providing a more natural set of partner molecules as no structural modifications were made except for the optical isomer relationship. To evaluate the D-core streptavidin capabilities, researchers used a biotin–streptavidin sandwich chemiluminescent enzyme immunoassay measuring thyroid stimulating hormone (TSH). They conjugated L-biotin to anti-TSH antibodies to detect the TSH in the sample. Increasing concentrations of D-biotin were spiked into patient samples to evaluate the effect of interference. The results show that the natural core streptavidin system displayed a signal decrease dependent on the amount of D-biotin added, and the genetically modified D-core streptavidin system did not have any significant signal change. By exploiting the importance of chirality in binding interactions, researchers were able to not only decrease the effect of exogenous D-biotin interference on a TSH thyroid hormone assay, but were also able to maintain the high binding affinity found in the native biotin–streptavidin system [63].

### 5.3. Direct Biotin Measurement

Incorporating a biotin immunoassay that screens for high doses of biotin into a patient's diagnostic panel is highly desirable. This provides the clinician with a tangible result for a straightforward indication of biotin interference as a possible reason for an erroneous result, and eliminates the need for additional methods to prove interference with biotin. Currently, there are manual commercial ELISA kits that can detect biotin concentrations as little as 0.400 ng/mL. The development of lateral flow formats and point-of-care tests

would be highly desirable for the rapid detection of biotin in an emergency clinical setting. Lab-based procedures such as High-Pressure Liquid-Chromatography Tandem Mass Spectroscopy (HPLC–MS) can also be used to measure biotin concentration, although it would not be suitable for processing a large number of public samples [29].

*5.4. Biotin Depletion Protocols*

Depletion protocols allow for the removal of the biotin in the sample before testing. By adding streptavidin-coated agarose beads to the sample, followed by incubation and centrifugation, biotin interference is revealed if the test result is significantly different before and after the removal [40]. A major advantage of using streptavidin-coated beads is the ease of separation of the free biotin in the sample, with well-established procedures and commercial availability [29]. However, depletion protocols generally need more thorough evaluation prior to implementation [40]. Dilution of the patient sample can also lower the level of biotin to help mitigate interference. As the sample is diluted, the concentration of both biotin and the analyte decreases and thus the interference decreases. Nonlinearity is an indication of biotin interference [29]. While the protocols may confirm the presence of biotin, test results using these methods should not be reported. They should be only used as a guide for the physician to request that the patient discontinue biotin supplementation and return for another blood draw at a later date [40].

The depletion approach is demonstrated in the study by Schrapp et al. to attempt to reduce biotin interference in a cardiac troponin T immunoassay by pre-treating patient plasma samples with streptavidin-coated microparticles. This highly sensitive immunoassay of Roche Diagnostics (cTnT-hs) is known to have interference with biotin [59]. Sandwiched complexes containing troponin T compete with the exogenous biotin in the sample for immobilization to streptavidin. Researchers took four native patient plasma samples with troponin concentrations that spanned the assay quantitative range (18, 59, 201, and 6423 ng/L) and spiked them with increasing concentrations of biotin (50, 100, 500, and 1000 µg/L). Researchers observed a mean negative bias of troponin concentrations of 23.5%, 55.8%, 96.7%, and 98.2% at 50, 100, 500, and 1000 µg/L of spiked biotin, respectively, demonstrating that the interference is independent of the initial troponin concentration. Biotin had such an impact on the results that the spiking events of 500 and 1000 µg/L decreased the troponin values of 18 and 59 ng/L samples lower than the limit of blank (3 ng/L) and decreased the troponin values below the 99th percentile (14 ng/L) for the 18, 59, and 201 ng/L samples. Researchers then followed a neutralization protocol in which a 1000 µg/L biotin spiked sample with an initial troponin T concentration of 59 ng/L was incubated with the same streptavidin-coated microparticles used in the troponin T assay for varying incubation times of 0, 15, 30, 45, and 60 min. The results show a complete suppression of interference across all incubation times (mean value 56.8 ng/L, 1.31% CV). Emergency biomarkers like troponin require quick analysis as time highly impacts patient treatment. While there was a complete suppression of interference with no incubation time, the neutralization process showed a dilution of 4.5–9.6% of the initial troponin value, altering the analytical performance of the assay. If a patient is suspect of myocardial infarction, a second assay one hour later upon initial admission is performed, known as the H0/H1 protocol. This protocol requires a high level of analytical quality as differences of 5 ng/L between the hour 0 and hour 1 samples determine diagnosis of myocardial infarction. This type of protocol should be avoided in this instance [59].

Since the publication of that article, Roche has now redesigned its assay to provide a greater tolerance to biotin interference. It used the same design format but added a monoclonal antibody against free biotin to bind and neutralize the free biotin in the sample. This antibody is specific only to the free biotin and does not bind to the biotinylated antibody/antigen conjugate. This novel method, however, is expected to incur high costs and could not completely eliminate biotin from the sample [29]. Anti-biotin antibodies have the same affinity to biotin as streptavidin when biotin is attached to a bovine serum albumin; however, anti-biotin antibodies bind to the free biotin in solution with a much

lower affinity. This could be explained by the dependence on the biotin accessibility when it is attached to a macromolecule and high biotin accessibility when it is free in the solution. Avidin's binding sites are located in a depression near the end of the β-barrels, making it more difficult to bind with biotin when the biotin is conjugated to a macromolecule. The anti-biotin antibody-binding sites for biotin are located on the end of the Fab segments, making the binding affinity less dependent on the biotin accessibility. In addition, the Fab segments are connected to a hinge, allowing for flexibility to adjust the spacing and orientation [34].

*5.5. Biotin Washout Period*

A washout period is defined as the amount of time between treatments necessary to prevent misinterpretation of a diagnosis being influenced by prior therapies [76]. Biotin washout periods allow for the elimination of high concentrations of free biotin from supplementation and are perhaps the easiest and best way to prevent interference. Biotin is non-mutagenic, non-toxic, and 100% bioavailable as biotin absorption and excretion are rapid. Only a limited amount of biotin catabolism occurs and the excess biotin is discharged in the urine. Biotin is absorbed from the gastrointestinal tract and reaches its peak levels in the blood after about 2 h. However, peak biotin levels vary between patients, depending on renal function, age, sex, and dosing frequency [29]. In a study by Grimsey et al., 54 apparently healthy volunteers ingested biotin supplements of increasing concentrations of 5, 10, and 20 mg once daily for 5 consecutive days. Blood samples were collected prior to biotin intake on days 1, 2, and 7, and at 1, 3, 6, 8, and 12 h post intake on days 3 and 7. The median peak serum biotin concentrations 1–3 h after the ingestion of 5, 10, and 20 mg of biotin were 41 (10–73) ng/mL, 91 (53–141) ng/mL, and 184 (80–355) ng/mL, respectively. The duration required for serum biotin concentrations to fall to 10 ng/mL and 30 ng/mL ranged from 1.5 to 73 h and 0 to 31 h, respectively [77]. Peak biotin levels in subjects with a normal dietary intake of biotin without supplementation are less than 0.5 ng/mL, while multiple sclerosis (MS) patients have a peak biotin level of 1160 ng/mL after taking a megadose of 300 mg. It is important to mention that the reported value of plasma biotin in the literature only reflects the concentration of free biotin (81%). Approximately 12% of biotin is also covalently bound to proteins and another 7% is reversely bound to proteins. For patients consuming biotin supplements in concentrations of up to 10 mg/day, a washout period of roughly 8–10 h is considered sufficient to allow biotin levels to return to near-physiological levels and circumvent interference. In contrast, patients consuming biotin supplements in concentrations of 300 mg/day have a considerably longer washout period of up to 73 h. Biotin washout periods are also impacted by impaired renal function, potentially resulting in longer durations that can span several days or weeks. Since the washout period differs widely among patients, biotin supplementation should cease at least 48 h before a blood draw in agreement with the guidelines issues by Mayo Clinic [29]. However, additional research is necessary to evaluate the biotin washout period for patients who continuously take biotin supplements over a long period of time.

Immunoassays can also be affected by a wide range of other interferences, including hemolysis, lipemia, icterus, heterophile antibodies, drugs, and other endogenous substances [40]. Biotin metabolites such as methyl ketone, bisnorbiotin, biotin sulfone, and bisnorbiotin $_{D,L}$-sulfoxide can also interfere as they bind to the biotin-binding sites of avidins. Two other known metabolites are tetranorbiotin $_{D,L}$-sulfoxide and tetranorbiotin and do not bind to avidins. Interference with biotin metabolites has not been extensively investigated, as it requires intensive efforts and resources. This is worth noting considering biotin accounts for only half of the total avidin-binding substances in human serum; thus, future studies are necessary to investigate the potential interference of biotin metabolites [29]. Due to the many different potential interferences, a systematic approach is necessary when evaluating discrepant results rather than focusing on a particular interferent [40].

### 6. Anti-(Strept)avidin Antibody Interference

While biotin interference has been a well-characterized and known issue among providers and clinicians, interference with anti-(strept)avidin antibodies has garnered attention as there have been numerous recent reports detailing such interferences [78]. The presence of anti-(strept)avidin antibodies in serum can cause the same bidirectional interference as biotin: falsely lowered results in sandwich assays and falsely elevated results in competitive assays. Unlike biotin, this interference is endogenous and thus expected to be present in circulation for a longer period of time than exogenous biotin [60]. Interestingly, interference with anti-avidin antibodies not only inhibits the binding of biotin, but also displaces the bound ligand. Antibody-induced conformational alteration of avidin destabilizes the ligand-binding characteristics at its interior. It is unusual that a high-affinity interaction can be curtailed allosterically due to antibodies with much lower affinity ($K_D$ of $\sim10^{-9}$–$10^{-10}$ M). This demonstrates the profound conformational change in avidin upon binding to its antibody, which significantly reduces its ligand affinity [79].

Although research has confirmed that humans possess natural antibodies against avidin, there is limited information regarding the properties of the immunoglobulins [80] and the mechanisms behind their formation [60]. In an effort to understand the factors that drive anti-avidin antibody production, questions about the biological role and function of avidins are raised. It has been suggested that avidins function in nature as antimicrobial agents by depleting biotin [5]. In oviparous animals, avidins are produced in oviductal tissues in response to bacterial, viral, or environmental stress. It is theorized that these proteins help protect eggs from microbes in the surrounding environment. In bacteria, researchers hypothesize that avidins are used to compete with other organisms by defending their environmental niche, invading hosts, and inducing pathogenicity. These bacteria have been found among a diverse range of ecological niches representing alternative lifestyles, which suggests that avidins are not exclusively advantageous to a single environment or biological role. Bacterial avidins have been associated with both soil and aquatic environments, and have differing symbiotic or pathogenic relationships with their hosts [46]. *Streptomyces avidinii*, the bacterium that produces streptavidin, and *Bradyrhizobium japonicum*, the bacterium that produces bradavidin, are both found in soil [60,81] and inhibit bacterial proliferation by acting as biotin scavengers [45]. Bradavidin has a known symbiotic relationship with soybean plants in defending against microbial infection [53]. Although streptavidin is extensively used over its other avidin analogues in biotechnology, there is a notable lack in the literature concerning *Streptomyces avidinii*. Specific information about the bacterium is scarce. Perhaps a comprehensive analysis of *Streptomyces avidinii* can reveal the true role of streptavidin in nature and provide a better understanding of the production of its antibodies in humans.

Immunoassay interference with anti-streptavidin antibodies (ASAs) is on the rise as recent literature has documented several new cases. An investigation to determine the frequency of ASAs in the Norwegian population analyzed 42 serum samples with discordant thyroid function tests (TFTs). Interference profiles in 34 of the 42 (81%) serum samples were attributed to the presence of endogenous ASAs [61]. Another study investigated falsely elevated results of anti-CPP IgG antibodies, a marker used for the diagnosis of rheumatoid arthritis, caused by the interference of ASAs in a laboratory in Belgium. Here, 1000 serum samples from ambulatory patients, 286 serums samples from patients for which anti-CPP measurement was requested, and 89 serum samples from patients who had previously given a positive anti-CPP result were evaluated for the presence of ASAs. Out of the 1375 patients, 8 (0.6%) had falsely elevated results determined by ASA interference [82]. A researcher from this study launched another investigation to assess the prevalence of these antibodies. ASA IgG and IgM concentrations were measured in 500 random patient samples and 2 patient samples with a known ASA interference. Out of the 500 random samples, 8 (1.6%) samples exceeded the calculated IgM cut-off value. The 2 identified cases of ASA interference had higher IgM concentrations that stood out among the other 8. These 8 samples could be representative of the unknown prevalence

of ASAs in humans. Immunoblot patterns between the cases and controls did not did not have any differences, indicating that there may be a general presence of ASAs in the population [83]. The sample selection in these studies was vastly different and thus global prevalence cannot be addressed using these investigations alone. Because instances of observed ASA interference range across the literature, the potential for undetected ASA interference should be addressed. Since the interference of ASAs mimic biotin interference, there is a possibility that erroneous results are being mistakenly attributed as biotin-related and ASA interference goes undetected. Further characterization of this interference is necessary to develop methods for distinction between interferences of biotin, ASAs, and antibodies against other avidin analogues.

The circumstances that lead to antibody production against streptavidin in humans are unknown, although a continuum of anti-streptavidin reactivity is found in the general population. It is hypothesized that people may develop an immunological reaction to streptavidin due to exposure to this soil bacteria in their outdoor activities [61]. In vivo studies focusing on avidin–biotin application for drug delivery, imaging, and cancer treatments have documented that avidin and streptavidin can elicit immune responses in humans. Streptavidin and related avidins have been previously used in clinical trials as drug carriers, but resulted in difficult cellular uptake, low intracellular delivery, and high immunogenicity. The presence of an Arg-Tyr-Asp sequence in streptavidin is responsible for unspecific interactions with cell surface receptors and immunogenicity. Avidin's basic nature and the presence of carbohydrate moieties are responsible for unspecific interactions with sugar-binding proteins, as well as DNA [45]. With growing application of the avidin–biotin system in cancer treatments, particularly for radioimmunotherapy and as a means of blood–brain barrier drug delivery, more avidin-like proteins are being discovered or engineered to circumvent the high immunogenicity seen in avidin and streptavidin [80]. Researchers Kawato et al. found that immunogenicity can be reduced using site-directed mutagenesis of streptavidin. Six amino acids at solvent-exposed charged and aromatic residues proposed to be involved in its immune reaction were substituted, creating the streptavidin mutant LISA-314. This streptavidin mutant displayed low immunogenicity without impairing biotin binding and thermal stability [84].

Research has suggested that the antibodies against avidin in human serum allow humans to tolerate avidin to a certain extent [85]. The avidin tetramer remains stable and active under relatively extreme conditions and is resistant to most degradative enzymes, which may explain its ability to produce long-term immunoglobulins. These antibodies may be responsible for the rapid elimination of avidin from the circulatory system [80]. Avidin clearance from circulating blood and tissues is considerably faster than streptavidin clearance. A study by Schechter et al. evaluated the clearance and tissue distribution of radioactively labeled avidin and streptavidin injected intravenously into mice. The presence of avidin and streptavidin were monitored over a period of 96 h. At 6 h post injection, avidin was present in concentrations of 0.1–1.8% in the heart, lungs, muscle, bones and 3.2–9% in the liver, spleen, and kidneys. Blood clearance was rapid, decreasing down to 1% at 20 min after injection. The radioactivity in most organs decreased to below 1% after 24 h and was undetectable at 96 h. Streptavidin retention was noticeably higher. Streptavidin was present in concentrations of 6–20% in the circulating blood and most organs during the 48 h period post injection. Even after 96 h, radioactivity was present in concentrations of 2–6% in the heart, lungs, liver, and spleen. The streptavidin concentrations in the kidneys 48 h post injection were higher at 18–22% and decreased to 10% at 96 h. Levels in the blood after 20 min were 42% and decreased with an approximate half-life of 24 h. Notably, high doses of exogenous biotin did not affect the streptavidin distribution, but caused an 2–7 fold increase in the retention of avidin in some of the organs, especially the liver due to its carbohydrate side chain [86].

Although avidin and streptavidin have similar binding affinity to biotin, they are very different as evidenced by their amino acid composition, primary structures, and different patterns of tissue association and biodistribution. Additionally, they lack evolutionary

relatedness and do not display immunochemical cross-reactivity [86]. In a study determining the extent of the immunological cross-reactivities of avidin and streptavidin, the core antigenic sequence of avidin can be recognized using both anti-avidin and anti-streptavidin antibodies, but the core antigenic sequence of streptavidin reacts only with streptavidin antibodies and cannot react with avidin antibodies [87]. The use of different avidin analogues may potentially mitigate interference with anti-avidin antibodies in regard to a decrease in immunogenicity. Evidence shows that bradavidin II displays a dramatically deceased immunogenic response in comparison to avidin and streptavidin [53]. Mutations have also been shown to produce lowered antigenic responses, thus reducing antibody production. Antigenic sequences can be engineered to prevent binding to anti-avidin antibodies and to extend the interaction of avidin-mediated therapy in vivo [80].

Since avidin and streptavidin antibodies are found naturally in human serum, it is important to evaluate their potential impact on interference in immunoassays. Understanding the mechanism behind their production will help identify better methods to effectively mitigate interference.

## 7. Conclusions

Since biotin–(strept)avidin-based immunoassays produce one of the strongest known covalent bonds, they have a tremendous potential for research efforts. Given the evidence that biotin supplementation is beneficial in the treatment of type II diabetes, obesity, and IBD, biotin supplementation is likely here to stay. As the popularity of biotin supplementation continues to rise, so does the instance of biotin interference. Biotin sales experienced a substantial 58% increase, rising from $219,599,798 in 2014 to $349,101,078 in 2018, not even including sales through online retailers [29]. Despite the observed benefits of biotin supplementation, its interference has the potential to be a global issue. Interference with biotin and anti-(strept)avidin antibodies affects all test methods using the biotin–(strept)avidin system, even those beyond immunoassays. Efforts are necessary to prioritize strategies and promote methods to further reduce interference.

**Author Contributions:** Conceptualization, A.H.A.B. and C.B.W., writing—original draft preparation, A.H.A.B.; writing—review and editing, A.H.A.B. and C.B.W. All authors have read and agreed to the published version of the manuscript.

**Funding:** This research received no external funding.

**Institutional Review Board Statement:** Not applicable.

**Informed Consent Statement:** Not applicable.

**Data Availability Statement:** No new data were created or analyzed in this study. Data sharing is not applicable to this review article.

**Conflicts of Interest:** Amy Balzer is an employee of Siemens Healthcare Diagnostics.

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
