# Peer review of "An Analysis of the Biotin–(Strept)avidin System in Immunoassays: Interference and Mitigation Strategies"

_cimb, doi:10.3390/cimb45110549_

Round 1

Reviewer 1 Report

Comments and Suggestions for Authors

The work is well organized. It represents a profound contribution in the field of immunology, especially clinical immunoanalysis and diagnostics. However, not enough molecular biology/immunology subjects are concerned with regard to the main topic of the manuscript. For example, whether it is possible to use any avidin analog (besides streptavidin) to avoid the influence of anti-streptavidin antibodies, via comparing avidin analog structures.

It remains unclear, what efforts besides the use of specific antibodies are or can be undertaken for the removal of excess biotin from patients' sera/plasma, to avoid the interference from free biotin. 

Is this biotin and anti-streptavidin antibody interference characteristic for all avidin analogs or for only some of them? 

No biotin serum concentration(s) is(are) described that can be critical for the correctness of biotin-streptavidin system-based immunoanalyses. If these concentrations differ for different test-systems, what do they depend on?  

Comments on the Quality of English Language

Several typing errors are present. 

Author Response

Please see attachment.  A marked up version is attached as supplementary file.  Please refer to the supplementary file when reading the reply to reviewers for line numbers corresponding to changes.

Reviewer 2 Report

Comments and Suggestions for Authors

The article delves into biotin-(strept)avidin interactions, highlighting their importance and application in immunoassays due to the solid non-covalent bonds they produce. The discussion discusses avidin's various properties and derivatives from multiple sources, such as the egg white of Gallus gallus and the bacterium Streptomyces avidin.

The article discusses various approaches to mitigating biotin interference:

Replacement of the biotin-(strept)avidin system.

Modifications of immunoassays to increase biotin tolerance.

Direct measurement of biotin concentration in a sample.

Protocols for removing biotin from samples.

Another essential aspect being investigated is the interference caused by anti-streptavidin antibodies, which may produce results similar to biotin interference.

The article concludes by emphasizing the importance of biotin supplementation, given its therapeutic benefits. However, as biotin's popularity increases, so does the incidence of interference, making it imperative to prioritize interference mitigation strategies.

In conclusion, the article offers a comprehensive view of the benefits and challenges of the biotin-(strept)avidin system. Shedding light on its common uses and potential pitfalls highlights the urgent need for further research and public awareness. This would benefit biotechnology professionals, healthcare providers, and even the general public who may consume biotin supplements. The in-depth discussion combined with a structured presentation makes this article essential reading for those interested in biotin-(strept)avidin interactions and their implications in immunoassays.

Author Response

(The authors gave the same response as above.)

Round 2

Reviewer 1 Report

Comments and Suggestions for Authors

No comments and suggestions.

Comments on the Quality of English Language

Please correct: "Haar und Haut", not "Haar unt Haut". 

Page 18, 2nd paragraph: "The circumstances that lead to antibody production against streptavidin in human are (NOT is) unknown ..."